

**Contrasting large fire regimes in the French Mediterranean**
Anne Ganteaume[1], Renaud Barbero[1]
[1]RECOVER-EMR, Irstea, Aix-en-Provence, France
*Correspondence to*: Anne Ganteaume (anne.ganteaume@irstea.fr)





## Abstract

In the French Mediterranean, large fires have significant socio-economic and environmental impacts. We used a long-term geo-referenced fire time series (1958-2017) to analyze spatio-temporal variations of large fires (LF; ≥100 ha) throughout a fire-prone area of this region. This area was impacted in some locations up to 5 or 6 times by recurrent LF and 21% of the total area burned by LF occurred on a surface that previously burned in the past. We found distinct patterns between the East and the West of the study area, the former experiencing fewer LF but of a larger extent compared to the latter, with an average time of occurrence between LF exceeding 4000 ha <7 years and >50 years, respectively. This longitudinal gradient in LF extent contrasts with what was expected from mean fire weather conditions strongly decreasing eastwards but is consistent with larger fuel cover in the East. The temporal variation of LF, featuring a sharp decrease in both frequency and burned area in the early 1990s, highlighted the efficiency of fire suppression and prevention, reinforced at that time. However, the LF outbreak in 2003 due to the exceptional heat wave remains of major concern in the context of climate change.

## 1 Introduction

It is now unanimously agreed that large fires have most significant socio-economic and environmental impacts, threatening or damaging infrastructures, ecosystems, and even costing human life, especially in the expanding wildland-urban interfaces (WUI) (Blanchi et al., 2014; Syphard et al., 2012; Syphard and Keeley 2015; Radeloff et al., 2018). However, the definitions of what can be considered as a large fire are numerous (Shvidenko and Nilsson, 2000; Stocks et al., 2002; Barbero et al., 2014a, Stavros et al., 2014; Nagy et al., 2018; Tedim et al., 2018), the size of such fires being arbitrary or statistically assessed (Moritz, 1997). Yet, as taking a fire-size threshold can minimize the importance of smaller fires in highly fragmented landscapes; this notion has also been approached focusing on the upper part of the burned area distribution (i.e. the largest 10% of fires occurring in a region according to Nagy et al., 2018). Albeit the choice of the cutoff remains highly subjective and variable from one study to another. Usually, large fires represent only a small proportion of the total number of fires but they typically account for the bulk of burned area in many regions throughout the world (Stocks et al., 2002; San Miguel-Ayanz et al., 2013; Stavros et al., 2014, Barbero et al., 2014a, 2014b, 2015; Ganteaume and Guerra, 2018) and contribute in fact to the trend and interannual variability in the total burned area.





41       Large fires and fire severity have increased over the past several decades across parts of the
globe (Kasischke and Turetsky, 2006; Pausas and Fernández-Muñoz, 2012; Dennison et al., 2014;
Stephens et al., 2014), these changes being attributed to a combination of climate change (Westerling
et al., 2006; Bradstock et al., 2009; Flannigan et al., 2009; Barbero et al., 2015; Abatzoglou and
Williams, 2016) and past fire suppression (McKenzie et al., 2004; Littell et al., 2009; Miller et al.,
2009). However, these patterns are not universal and some landscapes, mostly in southern Europe,
have not experienced such increases in large fires and even showed a decreasing trend since the 1990s
(San Miguel-Ayanz et al., 2013; Ruffault and Mouillot, 2015; Ganteaume and Guerra, 2018), with
conflicting signals found across parts of Portugal and Spain (Turco et al., 2016). This overall fire
reduction has been attributed to an increased effort in fire management and prevention after the large
fires in the 1980s (Turco et al., 2016; Fréjaville and Curt, 2017).

52       According to Sugihara et al. (2006), several characteristics of fire are used to define fire
regimes, ranging from temporal attributes, such as seasonality, fire return interval and fire rotation, to
spatial attributes, such as fire size and spatial complexity, and magnitude attributes, such as fire line
intensity, fire severity, and fire type. In Mediterranean areas, bottom-up drivers are generally thought
to strongly influence fire regimes. Indeed, ignitions are mainly due to human activities (negligence or
to arson) as seen in California (Syphard and Keeley, 2015; Kolden and Abatzoglou, 2018) or in
southeastern France (Ganteaume and Jappiot, 2013) where very few fires are started by lightning
strikes (Ganteaume et al., 2013). On the other hand, fuel structure and composition largely control fire
spread probabilities and, therefore, the location of the largest fires (Moreira et al., 2011; Ganteaume
and Jappiot, 2013; Duane et al., 2015; Fernandes et al., 2016). Human activities often modify the fuel
structure in the landscape (Moreira et al., 2011), as did agricultural land abandonment as a result of the
rural exodus allowing the build-up of large amount of fuels (Moreira et al., 2011, Pausas and
Fernández-Muñoz, 2012). Additionally, top-down drivers including strong weather gradients, can help
define areas where large fires are most likely to occur (Moritz et al., 2012; Ruffault et al., 2016). Large
fires in Mediterranean climate ecosystems are often enabled by episodes of severe fire weather of
varying duration, that can be generated by dry and hot winds as seen in California (Keeley and
Fotheringham, 2003; Moritz, 2003; Abatzoglou et al., 2013; Kolden and Abatzoglou, 2018), or cold
but dry wind as seen in southeastern France (Ruffault et al., 2016). Collectively, climatic factors
alongside ignition sources, fuels, but also terrain and suppression forces are thought to influence fire
spread (Dickson et al., 2006; Bajocco and Ricotta, 2008; Moreira et al., 2011; Ganteaume and Jappiot,
2013). Extensive work has therefore sought to target bottom-up and top-down factors thereby
controlling the variation of large fire activity, including i) meteorological factors or fire weather
indices (Riley et al., 2013; Dennison et al., 2014; Barbero et al., 2015; Bedia et al., 2015, 2018; Trigo
et al., 2016; Ruffault and Mouillot, 2017; Ruffault et al., 2016, 2018), ii) vegetation availability and
fuel moisture conditions (De Angelis et al., 2012; Dennison and Moritz, 2009) and iii) both weather





and vegetation combined (Koutsias et al., 2012) alongside human activities (Syphard and Keeley, 2015; Syphard et al., 2017; Nagy et al., 2018).

Some of the aforementioned factors were found to be non-stationarity in time. For instance, changes in fire suppression policy over the last few decades have induced sharp decreases in fires in some Mediterranean regions (Pezzatti et al., 2013, Moreno et al., 2014; Fréjaville and Curt, 2017; Ganteaume and Guerra, 2018), partially modifying the functional relationships linking fire to climate (Higuera et al., 2015; Fréjaville and Curt, 2017; Syphard et al., 2017), and thus, decreasing or increasing fire activity independently of the climate forcing (Hawbaker et al., 2013; Syphard et al., 2007).

As the European Mediterranean region, the Southeast of France is a highly populated area and is characterized by an extensive WUI. Fire prone areas along the Mediterranean coast have been extensively built up, reducing in some cases the availability of fuels but greatly increasing the probability of human-started fires (Ganteaume et al., 2013). The region includes plant communities well adapted to Mediterranean climate conditions that confer on this area a high fire risk. In this region, the largest fire on record reached 11 580 ha although most fires are generally smaller compared to other Mediterranean countries that have recently experienced larger fires such as Spain or Portugal. However, because of the high proportion of WUI, these large fires are of major concern, especially in the most populated parts, where most fires are also concentrated. Moreover, an increase in fire recurrence and a shortening of the period between fires were shown to impact vegetation structure, especially with the decrease in mature tree cover (Ganteaume et al. 2009), including the loss of resilience of *Pinus halepensis* stands (Eugenio et al. 2006).

Previous works in the Southeast of France were based on gridded fire data commencing from the mid-1970s (Ganteaume and Jappiot, 2013; Lahaye et al., 2014; Ruffault and Mouillot, 2015; Ruffault et al., 2016; Fréjaville and Curt, 2017; Ganteaume and Guerra, 2018; Lahaye et al., 2018). Here, we used longer time-series of georeferenced fires extending back to 1958 to identify both long-term trends and possible spatial patterns in large fire distribution, including fire recurrence, the time since the last fire and the mean time interval between fires. Finally, we sought to relate these spatio-temporal distributions of large fires to climate conditions and vegetation availability.

## 2 Material and Methods

### 2.1 Study Area

The study area (total surface area of 11 157 km$^2$) comprised two of the 15 French administrative districts that composed southeastern (SE) France and which are among the most impacted by fires in terms of fire frequency (i.e. number of fires) and burned area (Ganteaume and Jappiot, 2013; Ganteaume and Guerra, 2018). The western part is characterized by an extensive WUI where the



ignitions are the most frequent (47% of the total ignitions occurred in the WUI; Ganteaume and Long-
Fournel, 2015). Most large fires occur in summer but their cause is often unknown and when it is
known, these large fires are mainly due to arson (Ganteaume and Guerra, 2018).

115        The two parts of the study area (Fig. 1), located on a West-East gradient of the Mediterranean,
share most climate characteristics albeit the amount of annual precipitation increases eastwards
(Ruffault et al., 2017). These areas also differ in the structure of landscapes; forested massifs are larger
in the eastern zone while the proportion of WUI and the urbanization are higher in the western area
(respectively, 15% vs 7%, Ganteaume comm. pers., and 394 vs 174 inhabitants km$^{-2}$,
https://www.geoportail.gouv.fr), as well as in the main flammable fuel types, due to the nature of the
bedrock (acidic soils being mainly located in the East contrary to limestone-derived soils in the West).
All these differences are hypothesized to affect the spatio-temporal pattern of large fires.

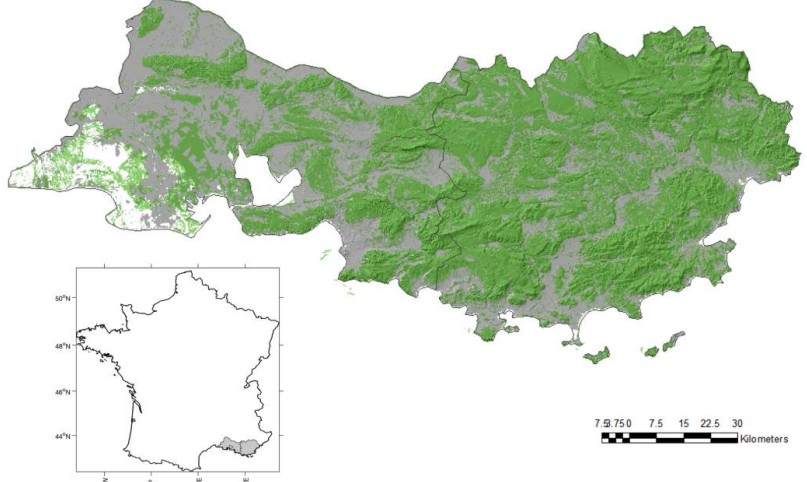


Figure 1: Map of the study area. Fuel cover in green was extracted from the "BD Forêt 2014" of the
National Geograhic Institute (https://www.geoportail.gouv.fr).

128        **2.2 Fire Data**

Large fires in SE France have already been studied in previous works using shorter time series based
on the gridded regional fire database Prométhée that recorded fires since 1973 (Fréjaville and Curt,
2015; Ruffault and Mouillot, 2017; Ruffault et al., 2018). However, this gridded data provides neither
the fire perimeter nor the temporal length needed to assess return periods in large fires. Here, we used
the georeferenced fire perimeter database of the Directions Départementales des Territoires et de la
Mer (DDTM Bouches du Rhône and Var) available from 1961 to 2017 in the western part and from





1958 to 2016 in the eastern part. We focused on large fires ≥ 100 ha (hereafter LF), representing only
28% of the total number of fires ≥1 ha (N=1277) but accounting for 94% of the total burned area.
Compared to large fires considered in other works (i.e. 200 ha in Canada according to Stocks et al.,
2002; 405 ha in the USA according to Dennison et al., 2014, 500 ha in Portugal according to Moreira
et al., 2011, and 1000 ha in Australia according to Bradstock et al., 2009), this detection threshold is
lower but within the range of thresholds used in other works in SE France ranging from 30 ha
(Ruffault and Mouillot, 2017) to 250 ha (Ruffault et al., 2017).

### 143    2.3    Climate and Land Cover Data

We computed the daily Fire Weather Index (FWI) from the Canadian Forest Fire Weather Index
system using daily surface meteorological variables at a 8-km spatial resolution from the quality-
controlled SAFRAN dataset providing minimum and maximum temperature, relative humidity,
precipitation and wind speed over France from 1959-2017 (Vidal et al., 2009, 2010, 2012). Although
the FWI was empirically calibrated for estimating whether atmospheric conditions and fuel moisture
content are prone to wildfire development in Canada (VanWagner, 1987), the FWI has already proven
useful in Mediterranean regions (Dimitrakopoulos et al., 2011; Fox et al., 2018; Lahaye et al., 2017).
Grid cells of the FWI lying within the study area were first averaged across the June-September season
and then averaged across all latitudes spanning the region of interest to form a longitudinal cross-
section of mean summer FWI conditions.
We extracted fuel cover data from the "BD Forêt 2014" of the National Geograhic Institute
(https://www.geoportail.gouv.fr) and regridded the data onto a 8-km spatial grid. The percentage of
land area covered by fuel was computed across all latitudes spanning the region of interest to form a
longitudinal cross-section as described above.

### 159    2.4    Spatial Analyses

We defined the LF regime in terms of the time interval since the last fire (LF ranging from "recent":
less than one decade to "ancient": more than four decades). LF recurrence was calculated by counting
the overlaps of LF polygons to quantify the number of times each location has been burned across the
period 1958-2017. For each LF, its georeferenced location and perimeter as well as the year of
occurrence were used to derive a fire return level in the western and eastern part of the study area, a
recurrence on a given location and the age of the last burned area.
Comparisons of means in burned areas due to LF were performed using a non-parametric
Mann-Whitney test and a Chi2 test was used to test the difference in number of LF between the two
parts of the study area.






### 2.5    Temporal Analyses

Monotonic trends in LF frequency and in burned area due to LF were assessed using the non-parametric Mann-Kendall test (Kendall, 1975) and a change point detection test (Standard Normal Homogeneity Test (SNHT); Alexandersson and Moberg, 1997) was used to identify potential abrupt changes in the time series.

We estimated LF return levels in the eastern and western part of the study area using the so-called block (here 1-year) maxima approach. We extracted the annual maximum LF size in both areas and selected the type of distribution that best fitted both series using the Akaike Information Criteria (AIC). In both areas, the gamma distribution was found to best describe the annual maximum LF size series. Using this distribution, the inverse cumulative distribution was calculated allowing the determination of the theoretical quantiles from which we derived the return levels (LF extent) associated to different LF return periods ranging from 5 to 100 years. Asymmetric confidence intervals were calculated using a resampling approach. This approach consists in creating new sub-samples from the original sample (75% of the original sample are extracted at random) using a bootstrapping process with replacement and then estimating a return level for each of the resampled data (N=1000). The resulting empirical distribution can then be used to derive the 95% confidence intervals from the resulting collection of estimates.

187

## 3    Results

### 3.1    Spatial Patterns of LF

In total, 353 LF occurrences were recorded in the study area between 1958 and 2017 (194 in the western part and 159 in the eastern part; Chi2=123.7 and p<0.0001) with, however, a higher burned area in the East nearly doubling the area burned in the West (respectively, 199 404 and 112 043 ha representing 3379.7 and 2000.8 ha burned per year; W=19306.5 and p<0.0001; Tab. 1). LF were responsible for most of the total burned area in the East (97%) as well as in the West (87%), which confirms the relevance of the fire-size threshold selected (100 ha).

Regarding the age distribution, LF were more frequent and burned the largest area (in the West only) between 1984 and 1975 (class 31-40 years; 25% to 29.6% of the occurrence from the West to the East and 27.2% of the total burned area in the former part). In contrast, the area burned by such fires was the largest before 1964 (class > 50 years; 28.2%; Tab. 2) in the East (this result was partly due to earlier beginning of the fire series in this part of the study area; i.e. 1958). In this latter part, recent LF were mainly located on the coast where the LF occurrence was the highest while in the


western part, the distribution of LF according to their age was more homogeneous (Fig. 2). Notice that
most LF growths were in the main wind direction blowing from Northwest.
A total surface area of 312 447 ha was burned during the period studied of which 21%
occurred on a surface that already burned in the past (Fig. 3), due to multiple overlaps in burned areas
by recurrent fires (i.e. LF occurrence on the same surface). LF recurrence occurred up to 5 times in the
West and up to 6 times in the East but represented only a small part of the recurrence (0.2% and 0.3%,
respectively; Tab. 3). In contrast, one LF and two recurrent LF were the most frequent patterns in the
western part of the study area (39.4 and 39.9% of the recurrence, respectively; Tab. 3) while, in the
East, most LF occurred only once (46.3%) on the same surface. The surface impacted by only one LF
represented 74.5% and 71.2% of the total area burned by LF in the West and the East, respectively
during the period studied and, as previously shown, the burned area involved in the highest recurrence
(area burned five and six times) was the lowest in both parts of the study area (0.005% and 0.008%,
respectively; Tab. 3).

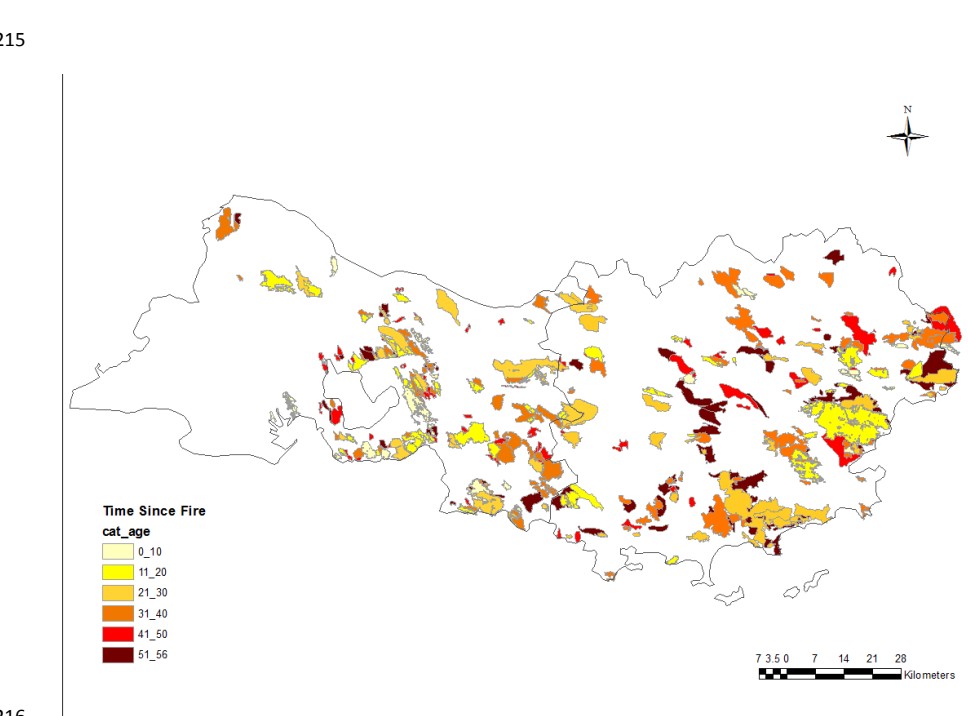


Figure 2: Time since the last LF (cat_age in years).


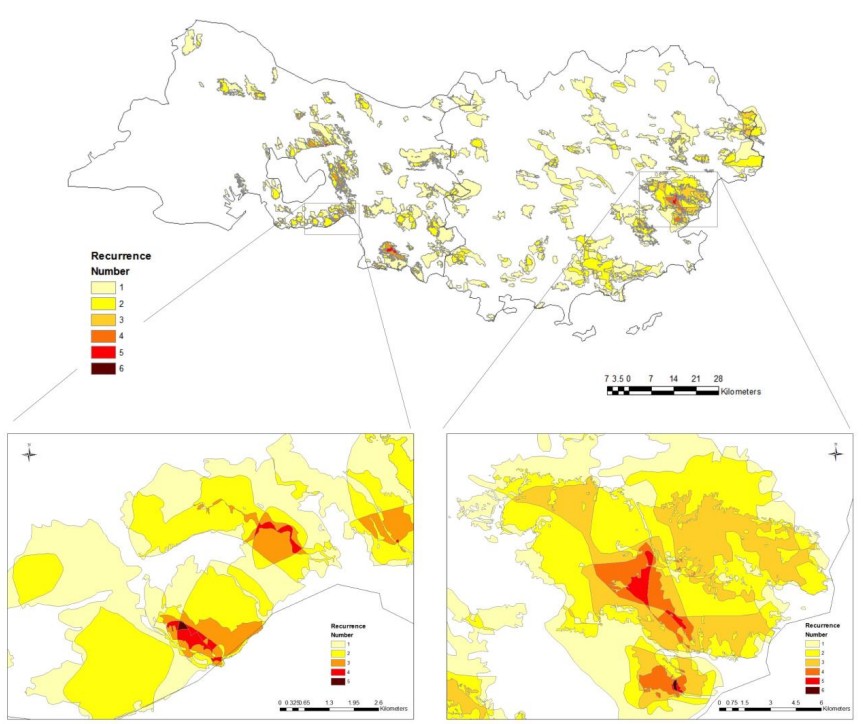


Figure 3: Fire recurrence on the 1961-2017 and 1958-2016 period in the western and eastern part, respectively.

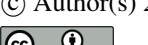



The mean LF extent varied along a longitudinal gradient, increasing from the West to the East
(Fig. 4 upper panel). This signal contrasts with the mean summer FWI gradient decreasing towards the
East but is consistent with the sharp increase in biomass cover towards the East (Fig. 4 lower panel).
This suggests that LF spread is not limited by climate conditions across the region but strongly fuel-
limited in the West, due to landscape fragmentation and the high proportion of WUI.

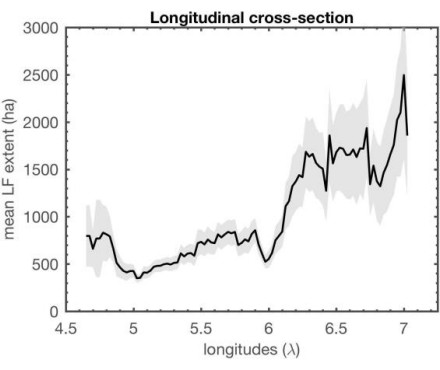

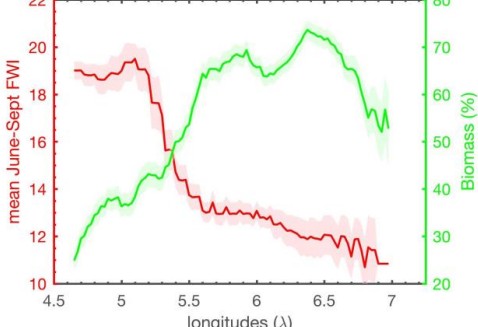


Figure 4: Top) Longitudinal cross-section of LF extent computed over 30-km sliding windows. The
95% confidence intervals were estimated using a bootstrapping approach. Bottom) Same as top panel
but for mean June-September FWI (in red) and the percent of fuel cover (in green).

**3.2    Temporal patterns of LF**
Figure 5 shows the annual LF frequency alongside area burned by LF in the entire region and in the
two parts of the study area separately. For both parts, 1979 and 1989 were the years presenting the
highest frequency of LF (respectively 11 for both years in the eastern area and 20 and 12 in the
western area). These years were also the most impacted in terms of area burned by LF in the western
area (respectively, 14 324 and 14 033 ha burned) as opposed to the eastern area more impacted in




1990 (24 920 ha burned). A significant change point in LF frequency as well as in burned area by LF
was detected in 1991 in agreement with previous findings (Fox et al., 2015) while it occurred around
1986 (Ruffault and Mouillot, 2015) in a slightly different area. This signal was especially evident in
the eastern part (Fig. 5c) while neither a change point nor a significant trend (p>0.05) were detected in
the western area for both LF metrics (Fig. 5b).

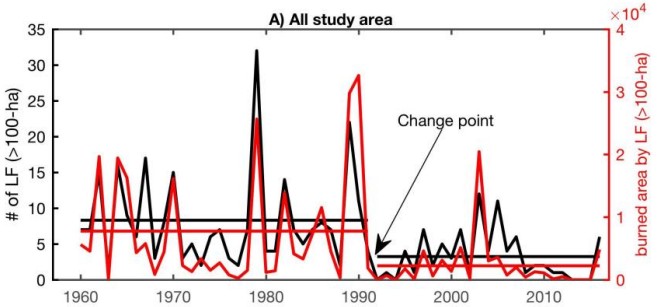

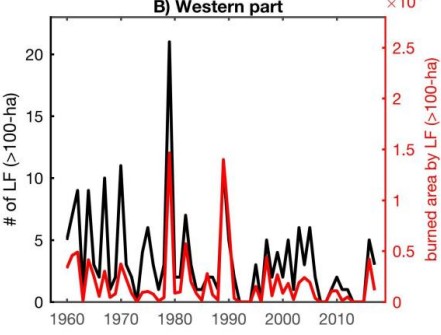
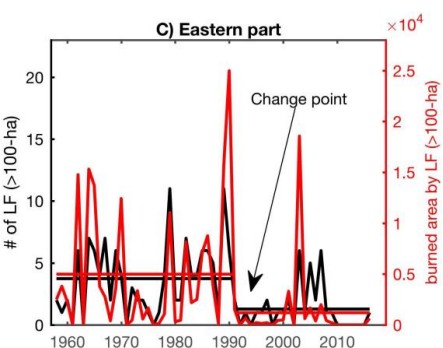


Figure 5: a) Annual number of LF (in black) and area burned by LF (in red) across the region.
Significant change points at the 5% confidence level according to a Standard Normal Homogeneity
Test (SNHT) in both metrics are indicated. Horizontal solid lines indicate the overall mean observed
before and after the change point. b) Same as a) but for the western part. c) Same as a) but for the
eastern part.

248         Figure 6 shows the annual maximum burned area in each part of the study area and the

Gamma distribution models that were found as the best fit to the data. Estimates of LF return intervals
show that a LF >4000 ha occurs on average every 7 years in the eastern zone and every 55 years in the
western area.




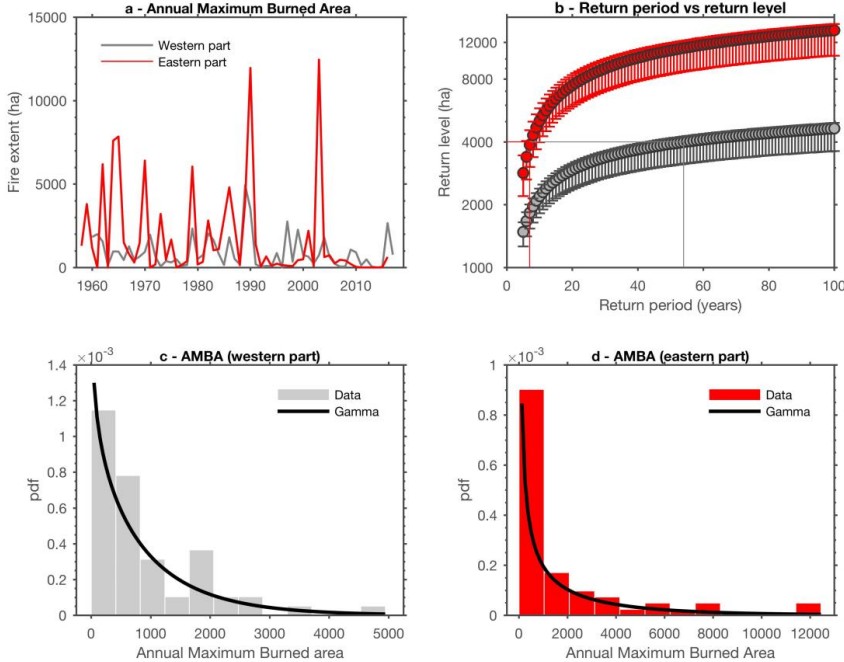


Figure 6: a) Time series of the annual maximum burned area in the western part (in gray) and in the eastern part (in red). b) Return levels in annual maximum burned area in the western part (in gray) and in the eastern part (in red) for different return periods ranging from 5 to 100 years. The 95% confidence intervals were estimated using a bootstrapping approach.

257

The correlation between mean June-September FWI and LF activity was computed over 31-year sliding windows (Fig. 7) and showed that much higher correlations in the western part than in the eastern part. The relationship strongly weakened after 1990-1991, a weakening that is more pronounced in the western part.

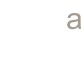
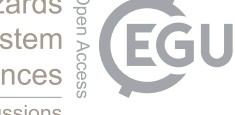


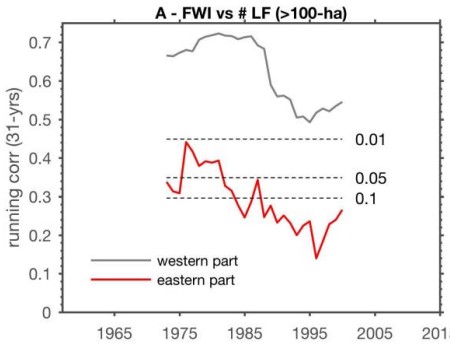
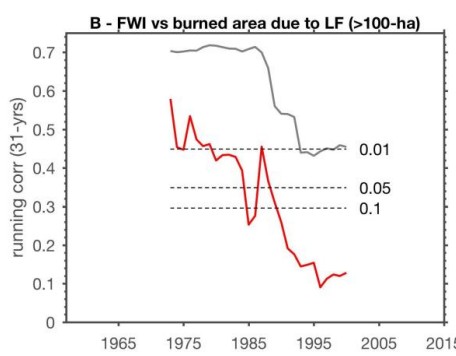

Figure 7: Sliding correlations on 31-year windows between mean June-September FWI and a) annual LF frequency and b) and annual burned area due to LF. The horizontal dashed lines indicate different significance levels of the Pearson correlations. Correlations are indicated for the middle of the sliding windows.

## 4    Discussion

Improving our understanding of large fire regimes is of upmost importance to fire prevention and management to mitigate their impacts. Here, we presented a comprehensive analysis of spatial and temporal patterns of LF in the French Mediterranean. To our knowledge, the fire database compiled and analyzed in this framework provides for the first time a detailed description of the LF regime recorded on geo-referenced long time series. Although previous works (Nagy et al., 2018) argued that using a single absolute size threshold to define a large fire was not a consistent indicator of ecological and economic risks across a large area (smaller fires may have stronger impacts than larger ones depending on the location), we opted for a fixed threshold of 100 ha as fires reaching or exceeding this size contributed to 94% of the total burned area and are likely to threaten ecosystems and/or the society. We, however, acknowledged that other metrics, such as fire intensity or fire damage (when available) may also provide additional insights on fire impacts (Tedim et al., 2018).

### 4.1    Spatial patterns of LF

We found that LF were larger but less frequent in the East compared to the West of the study area. Indeed, LF >4000 ha may occur within seven years in the East against 55 years in the West. In other words, LF are less probable in the east where fire ignitions are more limited but when an ignition does occur, the fire is likely to spread over larger areas. This longitudinal gradient is likely due to the variation in landscape fragmentation. Indeed, the western area presents a mosaic of wildlands interspersed with agricultural areas and WUI, LF being thereby concentrated in natural spaces less



extended than in the eastern part where large forested massifs mostly located on the coast allowed fire
spread. In contrast, LF were more frequent in the West where population density, the proportion of
WUI, and of infrastructures (railroads and roads) are the highest, this result agreeing with previous
works (Keane et al., 2008; La Puma, 2012; Alexandre et al., 2016; Nagy et al., 2018). Ruffault and
Mouillot (2017) showed that fuel fragmentation (i.e. due to a high proportion of WUI or road density)
was one of the most important factors limiting the occurrence of large fires in the French
Mediterranean agreeing with our results showing that the mean LF extent was more limited in the
West. However, our results also suggest that LF were slightly more frequent in the West despite the
fuel fragmentation. Fox et al. (2015) showed that, in an area located to the East of our study area,
neither WUI characteristics (despite the 60% increase between 1964 and 2009 in this area) nor
summer weather were major drivers of fire frequency and burned area, the climate control becoming
less important as the fire regime shifted to more frequent human-started fires (Zumbrunnen et al.,

300  2009).

In the western part, the most recent LF were mainly clustered along the coast while the more
ancient fires were located in the central and northern part. In contrast, LF were homogeneously
distributed in the East, regardless their age. LF recurrence (number of LF occurring on a same surface
during the period studied) was up to 5 times in the west and up to 6 times in the East. In the East, most
LF occurred only once on the same location and the largest areas were burned by ancient LF, while, in
the West, non-recurrent LF and especially two recurrent LF were the most frequent between 1975 and

307  1984.

Some recent studies across the Euro-Mediterranean countries emphasized that large fire
preferentially occured under specific synoptic patterns associated with high temperature (Pereira et al.,
2005; Trigo et al., 2013; Hernandez et al., 2015). In southern France, large fires were also facilitated
by wind events blowing from Northwest (Ruffault and Mouillot, 2015, 2017). The shapes of LF which
were more elongated in the wind direction in the western part support the results of Ruffault et al.
(2018) pinpointing that the main wind-driven large fires that had occurred in 2016 were located in the
western part while the main heat-driven large fires that occurred in 2003 were located in the East of
the area.

### 4.2    Temporal patterns of LF
The decreasing trend in both LF frequency and burned area observed over the last 6 decades is in
agreement with previous works (Ruffault et al., 2016; Turco et al., 2016) that highlighted a decrease in
fire activity across parts of southern Europe in response to an increased effort in fire suppression,
especially since the end of the 1990s in the French Mediterranean (Mouillot and Field, 2005). Indeed,
the region was highly impacted by fires during the 1970-1990 period and developed a thorough fire

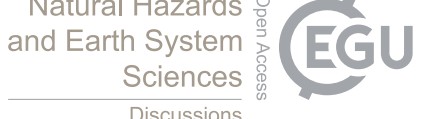

suppression and prevention system in the beginning of the 1990s, allocating more means for fire
management that allowed faster reactivity in case of fire start (the strategy became extinguishing the
fires at their initial stage by massive attack to prevent their spread). The decrease in both LF frequency
and burned area since 1991, especially evident in the eastern part of the region, is likely due to this
change in firefighting policy and fire prevention regulations (the fire suppression could be more
intense in the East because the fires were larger). This result was also highlighted in previous works
across the region (Curt and Fréjaville, 2017; Fox et al., 2015; Ruffault and Mouillot, 2015) as well as
in other countries (such as Switzerland; Pezzatti et al., 2013).
Climate projections suggest that atmospheric conditions conducive to large fire will increase in the
future, the warming and drying trends facilitating their probability of occurrence and their severity
(Stavros et al., 2014; Wang et al., 2015; Barbero et al., 2015), at least where fuel and ignitions are not
limiting. This trend towards more extreme fire weather conditions is likely to overcome prevention
efforts (Turco et al., 2016; Lahaye et al., 2018) in a region where expanding forests (Abadie et al.
2017) are increasing fuel loading and may offer opportunities for future fire spread.

## 5  Conclusions

This work, based on long-term geo-referenced large fire time series (1958-2017) analyzed spatio-
temporal variations of LF throughout one of the most impacted areas of the French Mediterranean. On
the whole, 21% of the total area burned by LF occurred on a surface that already burned in the past,
the region being impacted in some locations up to 5 or 6 times by recurrent LF. LF were less frequent
in the eastern part but larger than LF occurring in the West. This longitudinal gradient in LF extent,
with an average time of occurrence between LF exceeding 4000 ha <7 years and >50 years in the East
and the West respectively, contrasts with what we would expect from mean fire weather conditions
strongly decreasing eastwards but is consistent with larger fuel cover in the East. Recurrent LF
happened mostly in the WUI (especially in the West) and on the coast (especially in the East), while
non-recurrent LF were located inland in the East.
On the long-term, LF showed a clear decreasing trend in the early 1990s, mostly due to a
change in fire management policy thereby contributing to the weakening of the climate-fire
relationship. However, despites large means allocated to fire suppression, large fire outbreak is still
possible in SE France (such as in 2003 or 2016), indicating that specific weather conditions can
overwhelm the fire suppression efforts (Fernandes et al., 2016; Lahaye et al., 2018). A better
knowledge of the large fire regime is necessary to strengthen fire prevention by providing valuable
information on priority areas where recurrent LF are more likely to occur.





*Acknowledgements.* The authors wish to thank Adeline Bellet and Denis Morge, for preprocessing the
data with ArcGis. The authors also sincerely thank Aimee Mac Cormack for English revision.

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



588                                        TABLES

Table 1: Statistics on fires (≥ 1 ha) and LF (≥ 100 ha) in the study area

| Study area | Total number of fires | Total burned area (ha) | Number of LF: | % | Area burned by LF (ha) | % | Fire history (years) |
|---|---|---|---|---|---|---|---|
| West | 975 | 128 196 | 194 | 20 | 112 043 | 87 | 56 |
| East | 302 | 204 535 | 159 | 52 | 199 404 | 97 | 59 |
| Total | 1277 | 332 731 | 353 | 28 | 312 447 | 94 | |


Table 2: Percentages of area burned (relative to the total burned area) by LF and occurrence according
to the LF age classification in the two parts of the study area

| Class of age (years) | Western part | | Eastern part | |
|---|---|---|---|---|
| | Area burned by LF | Number LF | Area burned by LF | Number LF |
| 1-10 | 7.7% | 7.4% | 1.6% | 5.6% |
| 11-20 | 15.8% | 19.3% | 12.9% | 12.6% |
| 21-30 | 26.7% | 14.2% | 23.7% | 17.6% |
| 31-40 | 27.2% | 25% | 20.9% | 29.6% |
| 41-50 | 13.0% | 24.2% | 12.7% | 17% |
| >50 | 9.4% | 9.5% | 28.2% | 17.6% |


Table 3: Percentages of burned area (relative to the total burned area) affected by recurrent LF and
percentages of recurrence relative to the number of LF recurrence in the two parts of the study area

| | Western part | | Eastern part | |
|---|---|---|---|---|
| Recurrence | Area burned by | Recurrence | Area burned | Recurrence |



|  | recurrent LF |  | by recurrent LF |  |
|---|---|---|---|---|
| 1 | 74.5% | 39.4% | 71.2% | 46.3% |
| 2 | 20.3% | 39.9% | 22.3% | 34.7% |
| 3 | 4.5% | 16.6% | 5.5% | 13.1% |
| 4 | 0.7% | 3.9% | 0.8% | 4.1% |
| 5 | 0.005% | 0.2% | 0.2% | 1.5% |
| 6 | - | - | 0.008% | 0.3% |







598                                         FIGURE CAPTIONS


Figure 1: Map of the study area in southeastern France showing (i) the Numerical Terrain Model, (ii)
the forest fuel cover (extracted from the "BD Forêt 2014" of the National Geograhic Institute,
https://www.geoportail.gouv.fr) and (iii) the boundary of the two parts of the study area.

Figure 2: Map of the time since the last LF (cat_age in years).

Figure 3: Map of the LF recurrence on the 1961-2017 and 1958-2016 periods in the western and
eastern parts, respectively with zooms on the areas presenting the highest recurrence.

Figure 4: Top) Longitudinal cross-section of LF extent computed over 30-km sliding windows. The
95% confidence intervals were estimated using a bootstrapping approach.  Bottom) Same as top panel
but for mean June-September FWI (in red) and the percent of fuel cover (in green).

Figure 5: (a) Annual number of LF (in black) and area burned by LF (in red) across the region.
Significant change points at the 5% confidence level according to a Standard Normal Homogeneity
Test (SNHT) in both metrics are indicated. Horizontal solid lines indicate the overall mean observed
before and after the change point. (b) Same as (a) but for the western part. (c) Same as (a) but for the
eastern part.

Figure 6: (a) Time series of the annual maximum burned area in the western part (in gray) and in the
eastern part (in red). (b) Return levels in annual maximum burned area in the western part (in gray)
and in the eastern part (in red) for different return periods ranging from 5 to 100 years. The 95%
confidence intervals were estimated using a bootstrapping approach.

Figure 7: Sliding correlations on 31-year windows between mean June-September FWI and (a) annual
LF frequency and (b) and annual burned area due to LF. The horizontal dashed lines indicate different
significance levels of the Pearson correlations. Correlations are indicated for the middle of the sliding
windows.