# Peer review of "Contrasting large fire activity in the French Mediterranean"

_Natural Hazards and Earth System Sciences, 2018_

## Referee Comment (RC1) · Anonymous Referee #1 · 2 Nov 2018

The authors present a study that investigates and compares some of the characteristics of large fire regimes in two neighboring administrative regions of French Mediterranean Area. Based on a long-term (from late 1950's!) georeferenced local dataset of large fires (> 100 ha), they explore various options to quantify some spatial and temporal metrics and draw conclusions about the similarities and differences in LF regime between those two regions and their underlying drivers. Such issues related to the spatial temporal characteristics of LF regimes are of potential interest for the fire science community. Examine (or reexamine) some of these questions in Mediterranean France in the light of new elements brought in by a detailed dataset is therefore an attractive prospect. In general, this is a well-documented and well-written manuscript with clear language. However, I had a hard time understanding the main objectives of

this paper, or what were the author's purpose, which prevent from evaluating the added value of their study regarding the understanding of fire regime in this area. In its current form, the general impression is that this manuscript is a suite of more or less relevant and unrelated analyses that do not form or follow any guiding thread. There are many interesting ideas in this manuscript, as well as the use of an extensive and valuable fire dataset, but they are rather disconnected from each other and the general feeling is that the authors do not take full advantage of the potential of their dataset. I have attempted to summarize in the four points below my main concerns that should addressed for considering publication. (i) A lack of clear scientific hypotheses or research questions, (ii) improper interpretation and discussion of the outcomes of the analyses, (iii), questionable author's choices regarding the methodology and analyses and (iv) a lack description and of validation process of their dataset. I detail below these four main concerns.

**1 The research questions addressed in this study are not clearly stated. According to the title and some parts of the introduction (e.g. L52-56) the main goal of the authors is to investigate the "fire regime" of two different regions but that remains a very broad and undefined notion. Thus, and while the discussion is well written and very informative, a proper and clear scientific question is missing (see L98-104), which provides from clear conclusions of the article (see my point #2 below) and from a rationale choice of analyses (see my point #3 below). My opinion is that the authors should deeply revise their working hypotheses and focus their analyses around a few well-defined questions regarding fire regime characteristics.**

**2 The presentation of study's results appears mainly descriptive in some places, disconnected from the analyses in others, and a discussion of broader hypotheses, processes or wider implications is missing (probably because a clear research question is missing, see point #1 above). By way of example, the description of the main results and conclusions in the abstract (L13-23) is highly representative of the whole manuscript: the first sentences (L13-17) are very descriptive and no explanations are**

given by the authors about the consequence of these findings, the following ones (L18-21) are interesting but concerns two specific figures and by no means linked to the previous sentences or analyses and. The last sentence (L21-23) is very disconnected from the rest of the manuscript since none of the analyses presented by the authors specifically deal with the year 2003. More similar remarks could be made for the rest of the manuscript.

**3 Following points #1 and #2 above, some of the author's choices and rationale regarding data analyses are questionable. For instance, I could not understand what the information was brought by Table 2 and its description (L193-L201). It seems a less clear (and unnecessarily complex) representation of Figure 5. Figures 4, 6 and 7 are arguably the most interesting part, but unfortunately these are too superficially described and discussed. For instance, one might wonder about the impact of landscape transformation when studying such a long time period in figure 4 or about the meaning of the results from Figure 7 that is only very little discussed (L350-351). By contrast, Figure 5 is intensively discussed (L323-336) but does not bring much more information than previous studies on this topic. Finally, the authors mention several interesting ideas regarding for instance the study of fire shapes, but no specific analyses are made on this point, leaving their conclusions highly speculative (L308-315).**

**4. I agree the authors that long-term georeferenced fire dataset used in this duty is one major and significant novelty compared to previous studies. Yet this database is only to superficially described (L132-134) and no details are given about how data were collected and reported. Besides and while I understand the difficulty for a proper and full validation process of the author's fire dataset, I think that the study would really gain from a comparison of your dataset with other fire statistics products (such as Landsat) for fires shapes validation. Also, I was quite surprised by the number of fires (L136, N=1227 fires > 1 ha) that appear to be very low. For the period from 1973 to 2016 alone, the French official fire database (PROMETHEE, available at www.promethee.com) reports N=4561 fires > 1ha for the same two regions.**

---

## Referee Comment (RC2) · Anonymous Referee #2 · 18 Nov 2018

In this study the authors characterize the spatial and temporal patterns of large fire activity in a region of southern France where a longitudinal gradient in fire weather and land use conditions exists. This is a valuable addition to the literature, namely because it goes back in time more than usually available in Europe. However, in its present form, the manuscript falls short of fulfilling its potential, and I share most of the concerns expressed by reviewer #2, namely unclear objectives/research questions that forcefully lead to unfocused analysis and discussion. Additionally, I felt that many sentences are excessively referenced, which breaks the reading flow, and English expression could be significantly improved in order to make the ms. more appealing.

Specific comments

L14. "up to 5 or 6 times by recurrent LF" is redundant. Also, better to delete 5, as it is

not the maximum value.

L21-23. Rephrase, from my interpretation what is of concern is the future repetition of 2003-type events.

L32-33. The sentence is confusing, rephrase.

L39. I would say that is more "determine" than "contribute".

L50. Fire management includes fire prevention.

L79. Stationary, not stationarity.

L108-109. Comprised and composed should be in the present tense.

L119. It should be "unpublished data" or "data on file" rather than "pers. comm.", because Ganteaume is one of the authors.

L137-140. I don't think this is needed. Just keep the final parte relative to France.

L146. If you are using daily maximum (or means of minimum and maximum?) values of weather variables you are not calculating the FWI indexes correctly, which are based on noon observations. Clarify this.

L153. Why mean values and not a more extreme value, e.g. the 90th percentile, as large fires are known to occur under more extreme weather conditions?

L154. Clarify what you mean by "fuel", i.e. which land cover types are excluded or included.

L161. What you refer to as "recurrence" is overwhelmingly used in the literature as "frequency" (number of times burnt / time). I would advise to do the replacement across the entire manuscript, as it much more informative to report n/t than the number of times burnt.

L164. Here it seems you are referring to "fire return interval", but then we find out in results that the variable is expressed in hectares. Be more precise regarding fire

return "level". "age of the last burned area" is time since fire or patch age at the time it burned?

L166. Comparisons between the 2 regions?

L190. Delete "occurrences"

L301-307. This paragraph does not discuss results, presents them.

---

## Author Response (AR1)

The authors present a study that investigates and compares some of the characteristics of large fire regimes in two neighboring administrative regions of French Mediterranean Area. Based on a long-term (from late 1950's!) georeferenced local dataset of large fires (> 100 ha), they explore various options to quantify some spatial and temporal metrics and draw conclusions about the similarities and differences in LF regime between those two regions and their underlying drivers. Such issues related to the spatial temporal characteristics of LF regimes are of potential interest for the fire science community. Examine (or reexamine) some of these questions in Mediterranean France in the light of new elements brought in by a detailed dataset is therefore an attractive prospect. In general, this is a well-documented and well-written manuscript with clear language. However, I had a hard time understanding the main objectives of this paper, or what were the author's purpose, which prevent from evaluating the added value of their study regarding the understanding of fire regime in this area. In its current form, the general impression is that this manuscript is a suite of more or less relevant and unrelated analyses that do not form or follow any guiding thread. There are many interesting ideas in this manuscript, as well as the use of an extensive and valuable fire dataset, but they are rather disconnected from each other and the general feeling is that the authors do not take full advantage of the potential of their dataset. I have attempted to summarize in the four points below my main concerns that should addressed for considering publication. (i) A lack of clear scientific hypotheses or research questions, (ii) improper interpretation and discussion of the outcomes of the analyses, (iii), questionable author's choices regarding the methodology and analyses and (iv) a lack description and of validation process of their dataset. I detail below these four main concerns.

*Answer: We thank the reviewer for this thorough review and the suggestions that helped to improve the quality of the manuscript. We provide below a point-by-point response. We hope the research questions, objectives and the added value of the manuscript are now better stated.*

**1 The research questions addressed in this study are not clearly stated. According to the title and some parts of the introduction (e.g. L52-56) the main goal of the authors is to investigate the "fire regime" of two different regions but that remains a very broad and undefined notion. Thus, and while the discussion is well written and very informative, a proper and clear scientific question is missing (see L98-104), which provides from clear conclusions of the article (see my point #2 below) and from a rationale choice of analyses (see my point #3 below). My opinion is that the authors should deeply revise their working hypotheses and focus their analyses around a few well-defined questions regarding fire regime characteristics.**

*Answer:We agree with the reviewer. This part has been revised accordingly:*

*[...Previous works in the French Mediterranean were based on gridded fire data commencing*

*from the mid-1970s (e.g., Ruffault et al., 2016; Fréjaville and Curt, 2017; Ganteaume and*

*Guerra, 2018; Lahaye et al., 2018). Here, we used for the first time longer time-series of georeferenced fires extending back to 1958 and sought to examine both spatial and temporal distributions of large fires (>100 ha) across the French Mediterranean. More specifically, this paper has a three-fold objective. First, we sought to identify the locations associated with large fire recurrence and quantify the spatial extent of the region with reburns. Second, we sought to establish the mean fire extent and the fire return level along a longitudinal transect spanning the French Mediterranean and identify the possible role of climate conditions and fuel continuity in shaping this longitudinal gradient. This exploratory analysis may provide some insights on a fire aspect that was overlooked in previous studies. Finally, building on previous research, we sought to re-estimate trends in large fires across the region taking advantage of a fire record spanning almost six decades....]*

**2 The presentation of study's results appears mainly descriptive in some places, disconnected from the analyses in others, and a discussion of broader hypotheses, processes or wider implications is missing (probably because a clear research question is missing, see point #1 above). By way of example, the description of the main results and conclusions in the abstract (L13-23) is highly representative of the whole manuscript: the first sentences (L13-17) are very descriptive and no explanations are given by the authors about the consequence of these findings, the following ones (L18- 21) are interesting but concerns two specific figures and by no means linked to the previous sentences or analyses and. The last sentence (L21-23) is very disconnected from the rest of the manuscript since none of the analyses presented by the authors specifically deal with the year 2003. More similar remarks could be made for the rest of the manuscript.**

*Answer: The manuscript has been thoroughly revised according to these comments.*

**3 Following points #1 and #2 above, some of the author's choices and rationale regarding data analyses are questionable. For instance, I could not understand what the information was brought by Table 2 and its description (L193-L201). It seems a less clear (and unnecessarily complex) representation of Figure 5.**

*Answer: We have removed the unnecessary table and better commented fig. 5: [...A significant decline in annual LF frequency alongside area burned by LF was found across the region according to a Man-Kendall test (Fig. 5). This overall decline is consistent with a significant change point in both LF metrics in 1991 as shown in previous findings (Fox et al., 2015; Ruffault and Mouillot, 2015). This signal was especially evident in the eastern part (Fig. 5c) while neither a change point nor a significant trend (p>0.05) were detected in the western part for both LF metrics (Fig. 5b). We then examined how interannual correlations between mean June-September FWI and LF activity have changed over time across both regions (Fig. 5d). Higher correlations prevailed in the western part throughout the period but the relationships strongly weakened with time in both regions in agreement with previous findings (Ruffault and Mouillot, 2015), passing below significance levels across recent years...]*

Figures 4, 6 and 7 are arguably the most interesting part, but unfortunately these are too superficially described and discussed. For instance, one might wonder about the impact of landscape transformation when studying such a long time period in figure 4 or about the meaning of the results from Figure 7 that is only very little discussed (L350-351). By contrast, Figure 5 is intensively discussed (L323-336) but does not bring much more information than previous studies on this topic.

*Answer:This has been thoroughly revised according to these comments. Figure 7 has been combined to figure 5.The discussion has been divided into 2 new chapters: Spatial distribution of large fires and reburned areas and Long-term trends in large fires.*

Finally, the authors mention several interesting ideas regarding for instance the study of fire shapes, but no specific analyses are made on this point, leaving their conclusions highly speculative (L308-315).

*Answer: We added some points regarding the fire shapes but without being lengthy as this is beyond the scope of the paper: […Some recent studies across Euro-Mediterranean countries emphasized that large fire preferentially occurred under specific synoptic patterns associated with high temperature (Pereira et al., 2005; Trigo et al., 2013; Hernandez et al., 2015). In southern France, large fires were also facilitated by wind events blowing from Northwest (Ruffault and Mouillot, 2015, 2017). The shapes of LF which were more elongated in the wind direction in the western part support the results of Ruffault et al. (2018) pinpointing that the main wind-driven large fires that had occurred in 2016 were located in the western part while the main heat-driven large fires that occurred in 2003 were located in the East of the area. Taking into account other metrics describing the LF patch complexity (e.g. azimuthal angle or shape index) as in Laurent et al. (2018) could allow deriving additional information on the role of wind on their geometry or on the fraction of LF driven by wind…]*

**4. I agree the authors that long-term georeferenced fire dataset used in this duty is one major and significant novelty compared to previous studies. Yet this database is only to superficially described (L132-134) and no details are given about how data were collected and reported.**

*Answer:We tried to better described the database which was not an easy task as it is a governmental database and it is difficult to obtain a very detailed information! : […Here, we used the georeferenced fire perimeter database compiled by the Office National des Forêts (ONF) and Directions Départementales des Territoires et de la Mer (DDTM Bouches du Rhône and Var) available from 1961 to 2017 in the western part and from 1958 to 2016 in the eastern part of the study area. Fire perimeters were derived from aerial photography and remote sensing (the latter since 2016) and confirmed by ground truth targeting mostly fires larger than 10 ha in the earliest period. Approximate perimeters of older fire events (i.e., before 1990) have been corrected using aerial photos and Landsat satellite images when available (i.e. a more accurate delineation of fire perimeters adjustment were performed) (Faivre, 2011)…].*

Besides and while I understand the difficulty for a proper and full validation process of the author's fire dataset, I think that the study would really gain from a comparison of your dataset with other fire statistics products (such as Landsat) for fires shapes validation.

*Answer: We are not sure to understand this comment. Approximate fire contours that concerned older fires (i.e., before 1990) had been corrected using aerial photos and Landsat satellite images (i.e. delineation of unburned areas and fire boundaries adjustment were*

*performed). We provide this information in the material and methods section. If it is the comparison of all the fire shapes with Landsat images that is required, we think it is beyond the scope of this paper.*

Also, I was quite surprised by the number of fires (L136, N=1227 fires > 1 ha) that appear to be very low. For the period from 1973 to 2016 alone, the French official fire database (PROMETHEE, available at www.promethee.com) reports N=4561 fires > 1ha for the same two regions.

*Answer: Yes this is a good point raised by the reviewer. This number is indeed lower than in the Prométhée database (918 on the same period as that of Prométhée which began in 1973) as mainly fires >10 ha are preferentially targeted in the DDTM database. However, the total burned area did not differ that much: 219 878.8 ha in Prométhée and 215 163.6 ha in the DDTM dataset (as it is mainly driven by large fires, LF≥100ha) and regarding large fires, the DDTM database recorded a total of 237 LF which is very close of the 233 LF recorded in Prométhée. Same result regarding the burned area: 203 481.5 ha in the DDTM and 189 922.7ha in Prométhée. In conclusion, for both databases, the area burned by large fires represented between 88 and 92% of the total area which was the thing that matters most as the topic of this paper is related to large fires Moreover, other previous papers based on the Prométhée database gave the same conclusions.*

**Anonymous Referee #2**

In this study the authors characterize the spatial and temporal patterns of large fire activity in a region of southern France where a longitudinal gradient in fire weather and land use conditions exists. This is a valuable addition to the literature, namely because it goes back in time more than usually available in Europe. However, in its present form, the manuscript falls short of fulfilling its potential, and I share most of the concerns expressed by reviewer #2, namely unclear objectives/research questions that forcefully lead to unfocused analysis and discussion. Additionally, I felt that many sentences are excessively referenced, which breaks the reading flow, and English expression could be significantly improved in order to make the ms. more appealing.

*Answer: We thank the reviewer for this positive review. We tried to improve the readability of the manuscript and removed some references. We hope the manuscript is now more appealing.*

Specific comments

L14. "up to 5 or 6 times by recurrent LF" is redundant. Also, better to delete 5, as it is not the maximum value.
*Answer: This has been corrected.*

L21-23. Rephrase, from my interpretation what is of concern is the future repetition of 2003-type events.
*Answer: We removed this sentence following reviewer #1 suggestion.*

L32-33. The sentence is confusing, rephrase.
*Answer: This sentence has been corrected as requested.*

L39. I would say that is more "determine" than "contribute".
*Answer: This has been corrected.*

L50. Fire management includes fire prevention.
*Answer: This has been corrected.*

L79. Stationary, not stationarity.
*Answer: This has been corrected.*

L108-109. Comprised and composed should be in the present tense.
*Answer: This has been corrected.*

L119. It should be "unpublished data" or "data on file" rather than "pers. comm.", because Ganteaume is one of the authors.
*Answer: Yes indeed, this has been corrected.*

L137-140. I don't think this is needed. Just keep the final parte relative to France.
Answer: This has been corrected as requested.

L146. If you are using daily maximum (or means of minimum and maximum?) values of weather variables you are not calculating the FWI indexes correctly, which are based on noon observations. Clarify this.
*Answer: The reviewer is right. The FWI requires noon observations. Unfortunately, SAFRAN is a daily meteorological database and does not provide data at 1200 local time. We thus opted to use Tmax as a surrogate of noon temperature following prior analyses (e.g., Jolly et al. 2015; Abatzoglou et al., 2018). We clarified this point in section 2.3.*

L153. Why mean values and not a more extreme value, e.g. the 90th percentile, as large fires are known to occur under more extreme weather conditions?
*Answer: Both metrics (mean and extreme values based on percentiles) are relevant to track fire activity. Indeed, LF usually occur during periods of higher fire danger (e.g. consecutive days with FWI typically $>95^{th}$ percentile) but the amount of burned area over a season is also strongly correlated to mean seasonal FWI. We repeated Figure 4 using the $90^{th}$ percentile of each grid cell instead of the mean. As expected, results are highly similar, suggesting that the whole distribution of FWI (including both mean and extreme values) is shifting towards lower values as we move eastwards.*

[Figure]

Figure S1: Top) Longitudinal cross-section of LF extent computed over 30-km sliding windows. The 95% confidence intervals were estimated using a bootstrapping approach. Bottom) Same as top panel but for mean June-September 95th percentile FWI (in red) and the percent of fuel cover (in green).

L154. Clarify what you mean by "fuel", i.e. which land cover types are excluded or included. *Answer: This has been corrected as requested, the fuel cover types referred to the forest types.*

L161. What you refer to as "recurrence" is overwhelmingly used in the literature as "frequency" (number of times burnt / time). I would advise to do the replacement across the entire manuscript, as it much more informative to report n/t than the number of times burnt.

*Answer: Following the reviewer's suggestion, we replaced the total number of times burnt by the frequency expressed as number of times burnt during the entire period studied (56 years for the Western area and 58 years for the Eastern area).*

L164. Here it seems you are referring to "fire return interval", but then we find out in results that the variable is expressed in hectares. Be more precise regarding fire return "level".
*Answer: Figure 6b is actually showing the fire return interval for a given fire size (fire return level). We clarified this point in section 2.5 (Temporal analyses) and removed the reference to "fire return interval" from section 2.4 (Spatial analyses).*

"age of the last burned area" is time since fire or patch age at the time it burned?
*Answer: It is the time since the last fire*

L166. Comparisons between the 2 regions?
*Answer: Yes it is. We clarified this part.*

L190. Delete "occurrences"
*Answer: This has been corrected.*

L301-307. This paragraph does not discuss results, presents them.
*Answer: The results and discussion sections have been rewritten.*

---

## Author Response (AR2)

**Answers to reviewers' comments**

I read with great interest the new version of the Manuscript by Ganteaume and Barbero. The authors did a great job revising their manuscript and addressing reviewer's comment. I just have a few comments

- Throughout the manuscript the authors claim studying the effect of fuel continuity while, they are mixing the effect of both fuels quantify and continuity. The eastern Area having much more forested area than the western one, this might clearly increase the probability of large fires regardless of fuel distribution. You should consider normalizing fire statistics by forested area.

*Answer: We corrected the fire statistics presented in figure 4 as requested "[…] When normalizing by the biomass area (Fig. 4 top, right axis), the mean LF extent remains stable throughout the longitudinal gradient, except at the eastern-edge of the study area, where a sharp increase in LF extent per biomass surface unit was evident. This may be indicative of a stronger fuel connectivity at the eastern-edge, regardless of absolute biomass surface available to burn."*

- WUI is mentioned several times to explain fire patterns (e.g. L217, L278, L285, L288, L331) but none of these assertions are supported by data nor analyses. Please rephrase these sentences.

*Answer: We rephrased the sentences L278 and L331 according to this comment but not the other ones as we presented in the description of the study area, data on WUI showing that the eastern part of the study area presented lower proportion of WUI (7% vs 15%).*

-L193-194: It seems tautological
*Answer: We changed this sentence.*

-L195-205: This part is still rather descriptive. Statistical spatial analyses would be appropriate here to go further in those analyses and support your conclusions.
*Answer: We added statistical analyses (Chi2, anova) when relevant.*

-L290-293: Not sure I understand your point here
*Answer: The sentence was removed.*

-I would be also interesting to compare your results to the recent manuscript by Evin et al. (2018).
Evin, G., Curt, T., & Eckert, N. (2018). Has fire policy decreased the return period of the largest wildfire events in France? A Bayesian assessment based on extreme value theory. Natural Hazards and Earth System Sciences, 18(10), 2641-2651.
*Answer: Evin et al. (2018) examined return periods in burned area in a non-stationary context (before and after 1994) and over a slightly different region, making the comparison quite difficult. However, we added a couple sentences in the discussion:*

*"[…] It should be noted that return levels were estimated here under the assumption of a stationary context. Yet, the new fire policy that took place in the 1990s has been shown to reduce these return levels, albeit its effects on the largest fires were rather limited (Evin et al., 2018). Indeed, our estimates of 50-yr return levels in the eastern area lie within the confidence intervals of those observed in Evin et al. (2018) before and after the new fire policy. However, return levels in the West were much lower than those reported in Evin et al. (2018) across a larger region, highlighting the sensitivity of return levels to the spatial aggregation level of the data."*

-Figure 1: Please add a finer longitudinal scale to compare with the results from Figure 4

*Answer: Figure 1 has been improved according to this comment.*

[revised manuscript text omitted]